# Missing Plants Effects and Stand Correction Methods in *Coffea arabica* Progeny Experiments

**César Elias Botelho** [1,†], **Vinicius Teixeira Andrade** [1,†], **Juliana Costa de Rezende Abrahão** [1,*,†]
**and Flávia Maria Avelar Gonçalves** [2,†]

[1] Southern Minas Regional Unit, Minas Gerais Agricultural Research Agency (Epamig),
Lavras 37200-900, MG, Brazil; cesarbotelho@epamig.br (C.E.B.); vinicius.andrade@epamig.br (V.T.A.)
[2] Department of Biology, Federal University of Lavras (UFLA), Lavras 37200-900, MG, Brazil; avelar@ufla.br
[*] Correspondence: julianacosta@epamig.br
[†] These authors contributed equally to this work.

**Abstract:** Plant loss occasionally occurs in field experiments with coffee crops in experimental plots. In breeding programs, such loss can be harmful, especially when the corresponding statistical analyses are not consistent with the experimentally generated data. Herein, we analyzed productivity data to determine whether the compensatory effect occurs in coffee crops, evaluated the need to correct experimental failures, and identified the best stand correction method. We used productivity data from six harvests of eleven experiments with *Coffea arabica* plants. The experiments were implemented in a randomized block design, with four replications and six plants per plot. The following stand correction methods were evaluated: rule of three; Zuber; Vencovsky and Cruz covariance of the average or ideal stands; and Cruz, and the data were compared without correction adjustments. The most adequate correction methods were selected based on genetic variance, selective accuracy, and progeny ordering. The compensatory effect was evident from the analyzed data, with stand correction being evidenced as beneficial in progeny competition experiments. The best results were obtained for the covariance methods using average or ideal stands, followed by the method proposed by Cruz. The rule of three and Zuber method exhibited unsatisfactory results and are not recommended for stand correction in progeny competition experiments with coffee crops.

**Keywords:** *Coffea arabica*; stand correction; compensatory effect; unbalanced experiments

## 1. Introduction

Coffee crop productivity performance in Brazil is linked to the evolution of cultivation techniques and the use of improved cultivars [1]. Productivity gains have become significant over time and were achieved directly via cultivar selection with better production components and indirectly via lineage development for resistant and/or tolerant abilities against abiotic and biotic stresses.

In breeding programs, progeny evaluation is costly and time-consuming. Accordingly, experiments must be conducted with rigor and precision so that they have less errors and phenotypic differences representing the genotypes. The magnitude of experimental errors is directly related to agricultural experimentation and plant breeding success. Experimental accuracy is affected by several factors, such as soil heterogeneity, pests and diseases, plot size, design, replications, and number of plants per plot [2].

There are many questions regarding the methods used in coffee progeny evaluation, some of which have not yet been thoroughly researched. One such example is missing plants in the experimental plots, which poses a significant obstacle to accurate superior coffee progeny identification and reliable genetic parameter estimation [3]. Plant loss reduces experimental result precision and introduces bias, potentially leading to erroneous conclusions and suboptimal breeding decisions. For perennial crops, such as coffee, the

breeder must consider appropriate experimental techniques that minimize experimental errors. Losses caused by technique inadequacies can only be verified after several years and may reduce selection efficiency and delay progeny and/or new cultivar obtention. Whether factors are controllable or not, all factors that affect the error must be observed by breeders to increase selection process efficiency. Thus, determining the effect of missing plants in experimental plots and using methods to stand correction that can attenuate such effects are necessary to improve the accuracy of genetic parameter estimates.

Several stand correction approaches have been established. A remarkably simple method is presented by the rule of three (RT), which assumes proportionality between productivity and the number of plants in the experimental plot. However, such proportionality does not always occur, often resulting in a biased productivity value for the evaluated genotypes [4]. Zuber [5] proposed a second method (referred to as "Zb") to correct the RT error. However, this adjustment method also has limitations, wherein it does not consider the disposition of faults in the field, and its coefficient of compensation for lack of competition defaults to 0.3. Other methods, which have proven to be efficient in many cases, apply the analysis of covariance, wherein the final stand of the plot is assumed as a covariate. The methods proposed by Cruz [*apud* 4] and Vencovsky and Cruz [4] (referred to as "Cr" and "VC", respectively) have also been used efficiently.

The compensatory effect of the crop under plot failure is another matter of consideration, since the plants adjacent to the failures generally have higher productivity owing to less competition for water, nutrients, and light. Although the compensatory effect varies among different crops [6,7], there are no published reports regarding its occurrence in *Coffea spp.* plants and how it affects the accuracy of experiments with coffee crops. So, the stand correction methods must deal with reduction in plot bean yield due to lesser number of plants and the compensatory effect caused by the absence of one or more plants.

Therefore, the primary objective of this study was to determine the magnitude of the compensatory effect resulting from plant loss in *Coffea* crops. By quantifying the extent to which remaining plants compensate for the missing ones, we aimed to elucidate the implications of plant loss on coffee progeny evaluation accurately. Furthermore, we evaluated the necessity of correcting for the number of failures in coffee experiments and identified the most effective methods for stand correction. The knowledge gained from this study will assist breeders regarding the appropriate data analysis procedures to use for plant selection in the event of stand reduction due to management and/or climatic events, pests, and diseases.

## 2. Materials and Methods

Analyses were performed using production data (in kg plot$^{-1}$) from six harvests obtained from eleven *Coffea arabica* progeny testing experiments conducted by the Minas Gerais Agricultural Research Agency (Empresa de Pesquisa Agropecuária de Minas Gerais—EPAMIG). The experiments, which were performed in four cities in the state of Minas Gerais (Table 1), were conducted using a randomized block design, with four replications and six plants per plot.

F4 progenies from Icatu × Catimor and Mundo Novo × Catuaí crosses Mundo Novo, Catuaí, and Icatu lineages were analyzed (Table 1). Implementation and experimental conduct followed the technical recommendations commonly used for coffee cultivation in each region [8]. Phytosanitary practices were conducted according to the seasonal occurrence of pests and diseases. The individual plots were harvested annually between June and July. The coffee (coffee fruits of mixed maturity) volume obtained from each plot was converted into 60 kg bags of processed coffee per hectare (bags·ha$^{-1}$).

The compensatory effect was obtained by estimating the compensation coefficient (*a*) using the estimator $a = \frac{c}{\overline{Y}}$, where *c* is the linear regression coefficient obtained as proposed by Cruz [*apud* 4]; $\overline{Y}$ represents the average yield per plant obtained in the experiment. A compensation coefficient $\geq 1$ indicates a positive compensatory effect in the crop [4].

**Table 1.** List of experiments, progenies, locations, spacing, number of progenies (NP), percentage of failures (%), and significance (*p*-value) for the number of failures in the plots, general mean ($\overline{Y}\ldots$), and linear regression coefficient (b).

| Experiments | Progenies | Location | Spacing | NP | Failures (%) | *p*-Value | $\overline{Y}_{\ldots}$ | b |
|---|---|---|---|---|---|---|---|---|
| IxC-CA | Icatu × Catimor | Campos Altos | 3.5 × 0.5 | 30 | 6.80 | 0.16 | 19.30 | 1.11 |
| IxC-S | Icatu × Catimor | São Sebastião do Paraíso | 3.0 × 0.5 | 30 | 14.72 | 0.01 | | |
| MxC-TP | Mundo Novo × Catuaí | Três Pontas | 2.5 × 0.7 | 42 | 22.88 | 0.28 | 12.03 | 1.62 |
| MxC-CA | Mundo Novo × Catuaí | Campos Altos | 3.5 × 0.5 | 25 | 4.33 | 0.31 | 25.66 | −1.07 |
| MxC-CP | Mundo Novo × Catuaí | Capelinha | 3.5 × 0.5 | 25 | 12.83 | 0.89 | 11.11 | 0.88 |
| ICT-CA | Icatu | Campos Altos | 3.5 × 0.8 | 15 | 13.61 | 0.69 | 25.96 | 0.94 |
| ICT-CP | Icatu | Capelinha | 3.5 × 0.8 | 15 | 26.67 | 0.13 | 15.91 | 1.68 |
| MN-TP | Mundo Novo | Três Pontas | 3.0 × 1.0 | 35 | 10.33 | 0.40 | 22.22 | 1.99 |
| MN-CA | Mundo Novo | Campos Altos | 3.5 × 0.8 | 35 | 18.22 | 0.67 | 23.03 | 1.72 |
| MN-CP | Mundo Novo | Capelinha | 3.5 × 0.8 | 35 | 29.78 | 0.08 | 13.85 | 1.95 |
| CAT-CP | Catuaí | Capelinha | 3.5 × 0.5 | 20 | 46.25 | 0.01 | | |
| Mean | | | | | | | 19.15 | 1.20 |

Following the recommendation by Steel et al. [9], correction was not applied in case of significant differences, as determined by the F test, in the final number of plants per plot (stand) between treatments. In the absence of grain production data adjustments (AA), plant loss in the plot was disregarded (in this case, $Zij = Yij$). Otherwise, the productivity data were adjusted as a function of the variable stand using the following six procedures:

(i) RT method, using the expression $Zij = Yij(H/Xij)$, where *H* represents the ideal stand (in this case, six plants);

(ii) Zb method, using $Zij = Yij[H - a(H - Xij)]/Xij$, where *a* represents the compensation coefficient for the absence of competition (in this case, *a* = 0.3);

(iii) VC method, using $Zij = Yij[H - a(H - Xij)]/Xij$, where *a* represents the compensation coefficient estimated from experimental data, and *H* = ideal stand;

(iv) Analysis of covariance with the ideal stand as the covariate (hereinafter "ACI" method), using $Zij = Yij - b(Xij - H)$, where *b* represents the linear regression coefficient as a function of *Yij*, estimated according to the procedure described by Steel et al. [9];

(v) Analysis of covariance with the average stand as the covariate (hereinafter "ACA" method), using $Zij = Yij - b(Xij - ..X)$, where *b* represents the *Yij* residual regression coefficient, estimated according to the procedure described by Steel et al. [9], and ..*X* represents the average stand of the experiment;

(vi) Cr method, using $Zij = Yij(H/Xij) - c(H - Xij)$, where *c* represents the residual regression coefficient of the variable *Yij* (corrected using the RT method) as a function of the number of failures in the plot. In all the adjustment expressions above, *Zij* represents the corrected yield, and *Yij* represents the observed production in real plots/stands (*Xij*).

Analyses of the six crops were performed using Zij data. For that, the data were used in the following statistical model [9]: $Y_{ijq} = m + b_j + p_i + c_q + pb_{ij} + bc_{jq} + pc_{iq} + e_{ijq}$, where $Y_{ijq}$ represents the observation of the $ijq^{th}$ plot in block *j* of harvest *q*, which received progeny *i*; *m* represents the constant associated with all observations; $b_j$ is the fixed effect of the $j^{th}$ block; $p_i$ is the random effect of the $i^{th}$ progeny, where $p_i$~ NMV (0, $\sigma_p^2$); $c_q$ is the fixed effect of the $q^{th}$ harvest; $pb_{ij}$ is the random effect of the $ij^{th}$ progeny × block interaction, where $pb$~ NMV (0, $\sigma_{pb}^2$), representing the error *a*, which is the error at the plot level; $bc_{jq}$ is the fixed effect of the $jq^{th}$ block × harvest interaction; $pc_{iq}$ is the random effect of the $iq^{th}$ progeny × crop interaction, where $pc$ ~ NMV (0, $\sigma_{pc}^2$), representing the error *b* that is the error at the subplot level; and $e_{ijq}$ is the random effect of the experimental error associated with the observation of the $ijq^{th}$ plot, where $e$ ~ NMV (0, $\sigma_e^2$).

The mixed model approach with random treatments generally results in more homogeneous treatment means and different genetic treatment selection in scenarios with low genotypic variance compared to error variance, and when trials are non-orthogonal and unbalanced [10]. Thus, the empirical best linear unbiased prediction (E-BLUP = $\hat{h}^2(\overline{Y}_i - \overline{Y})$) value was estimated by considering each adjustment method and the uncorrected data [11]. Estimates of the experimental accuracy were obtained algebraically using the square root of the heritability at the mean progeny level with the following formulas:

$$\hat{h}^2 = \frac{\hat{\sigma}_p^2}{\hat{\sigma}_p^2 + \frac{\hat{\sigma}_{pc}^2}{q} + \frac{\hat{\sigma}_{pb}^2}{j} + \frac{\hat{\sigma}_{e}^2}{qj}}, \ and \ \hat{h} = \sqrt{\hat{h}^2},$$

where $\hat{h}^2$ represents the heritability at the mean progeny level; $\hat{h}$ represents the experimental accuracy; $\hat{\sigma}_p^2$ : progeny variance; $\hat{\sigma}_{ph}^2$ : variance of the interaction progeny $\times$ harvests; $\hat{\sigma}_{pb}^2$ : variance of the interaction progeny $\times$ block; $\hat{\sigma}_{e}^2$ : error variance; $j$: number of blocks; $q$: number of harvests.

The parameter estimates were used to verify which method would be the most efficient, including the mean $(\overline{Y}\ldots)$, standard deviation of the mean (s), $p$-value, and selective accuracy $r_{gg}$. The most effective methods for stand correction are those that result in less interference in the mean, smaller s- and $p$-values, and greater $r_{gg}$ estimates.

The E-BLUP estimates obtained with each correction method were used to estimate the index of coincidence (IC) for a selection intensity of 20%; that is, the proportion of superior progenies and/or lineages with the same behaviour for each method compared with the unadjusted data. The IC was estimated using the method proposed by Hamblin and Zimmermann [12], which considers the effect of chance, given by the following: $IC = \frac{A-C}{M-C} \times 100$, where $C$ represents the number of progenies and/or superior lineages selected as a result of chance (i.e., the number of selected progenies and/or superior lines in proportion to the selection intensity is assumed to coincide by chance); $A$ is the number of superior progenies selected, common to the different methods; and $M$ is the number of superior progenies and/or lineages selected with the AA method.

## 3. Results

The percentage of failures per experiment ranged between 4.33% and 46.25%, with the greatest loss observed in the Catuaí group lineage. In two cases (Icatu $\times$ Catimor progenies in São Sebastião do Paraíso and Catuaí lineages in Capelinha), significant differences among the progenies with respect to the plot stands were observed, indicating that the progeny effect influenced plant loss (Table 2).

**Table 2.** Compensation coefficient values (a) for experiments in each of the six harvests (H): H 01 to H 06.

| Experiment | H 01 | H 02 | H 03 | H 04 | H 05 | H 06 | Mean |
|---|---|---|---|---|---|---|---|
| IxC-CA | 0.89 | 1.38 | 1.61 | 1.21 | −0.08 | 1.72 | 1.12 |
| MxC-TP | 2.77 | 1.77 | 2.86 | 0.56 | 1.72 | 0.07 | 1.63 |
| MxC-CA | 0.94 | 1.017 | 2.73 | 2.14 | 0.80 | 0.74 | 1.39 |
| MxC-CP | −0.24 | 1.17 | −0.16 | 2.66 | 0.99 | 0.99 | 1.00 |
| ICT-CA | 0.39 | 1.11 | 0.42 | 1.16 | 0.14 | 1.75 | 0.83 |
| ICT-CP | 0.43 | 1.48 | 0.54 | 2.40 | 1.25 | 2.04 | 1.35 |
| MN-TP | −0.53 | 0.001 | 1.78 | 0.40 | 6.60 | 0.53 | 1.29 |
| MN-CA | 0.19 | 0.88 | 1.10 | 1.07 | 1.10 | 0.002 | 0.73 |
| MN-CP | −0.05 | 1.04 | −0.09 | 0.94 | 1.75 | 0.67 | 0.71 |
| Average | 0.53 | 1.09 | 1.37 | 1.39 | 1.59 | 0.95 | 1.15 |

Estimates of the mean linear regression coefficient (b), which reflects the coffee production response in the plot as a function of the change in the number of plants (stand), were generally expressive of the six harvests (Table 1). The Mundo Novo $\times$ Catuaí de Campos

Altos progeny experiment presented a negative b estimate, indicating that each failure resulted in a decrease of 1.07 kg in the plot. For the other cases, the b estimates ranged from 0.88 to 1.99, indicating an increase of 3.64–14.08% relative to the mean production of the plot.

Considering the existence of b, the compensation coefficient for lack of competition (a) was estimated for each experiment and year of harvest (Table 2). There were variations among the groups of progenies based on locations and, mainly, harvesting years. The greatest variation occurred in the Mundo Novo lineage experiment in Três Pontas, where estimates ranged from $-0.001$ (H 02) to 6.60 (H 05). In general, compensation within the same experiment increased over time until an equilibrium was attained. Coefficient comparisons between experiments revealed that the failures in the Mundo Novo × Catuaí experiment (conducted in Três Pontas) were the ones that influenced the productive response pattern of the plants the most ($=1.63$), since this experiment showed greater compensation for the lack of competition.

Having determined that failures did indeed change the productive response pattern of the coffee plants, it was necessary to apply methods that result in adequate selection. With this objective in mind, six methods for correcting *Coffea arabica* progeny production data were tested. Comparisons of the adjusted data with the original data (AA) indicated that the six correction methods used generally overestimated the means to different degrees (Table 3). The RT method provided the highest mean estimate, overestimating it by 18.72% in relation to the original data (AA). Likewise, the Zb method overestimated the mean of all trials by 15.56. In contrast, the best mean estimates were obtained with the VC and ACA methods, which overestimated the unadjusted mean by only 4.34% and 4.72%, respectively (Table 3).

**Table 3.** General means of the experiments according to the various data correction methods used and their percentage increment in relation to the unadjusted mean.

| Experiment * | AA | RT | Zb | VC | ACI | ACA | Cr |
|---|---|---|---|---|---|---|---|
| IxC-CA | 19.30 | 20.98 | 20.47 | 20.98 | 19.75 | 19.30 | 20.98 |
| MxC-TP | 12.30 | 18.65 | 18.38 | 17.81 | 18.18 | 17.75 | 17.98 |
| MxC-CA | 25.67 | 27.11 | 26.68 | 24.80 | 25.36 | 25.67 | 25.49 |
| MxC-CP | 11.12 | 13.07 | 12.48 | 11.03 | 11.79 | 11.12 | 11.57 |
| ICT-CA | 25.95 | 31.27 | 29.68 | 29.09 | 26.73 | 25.96 | 25.89 |
| ICT-CP | 15.91 | 23.99 | 21.57 | 10.20 | 18.59 | 18.59 | 15.91 |
| MN-TP | 22.22 | 23.35 | 23.01 | 22.53 | 22.69 | 22.22 | 22.87 |
| MN-CA | 26.03 | 30.63 | 29.25 | 28.06 | 27.31 | 26.03 | 26.47 |
| MN-CP | 13.85 | 19.29 | 17.66 | 15.34 | 16.75 | 13.85 | 16.15 |
| Average | 19.15 | 23.15 | 22.13 | 19.98 | 20.79 | 20.05 | 20.36 |
| Increment (%) | - | 18.72 | 15.57 | 4.34 | 8.59 | 4.72 | 6.32 |
| Standard deviation of the mean | 4.16 | 6.81 | 5.57 | 5.64 | 3.95 | 3.95 | 5.57 |

AA: absence of adjustment; RT: rule of three method; Zb: Zuber [5] method; VC: Vencovsky and Cruz [4] method; ACI: analysis of covariance based on the ideal stand; ACA: analysis of covariance based on the average stand; Cr: Cruz [*apud* 4] method. * IxC-CA: Icatu × Catimor in Campos Altos; MxC-TP: Mundo Novo × Catuaí in Três Pontas; MxC-CA: Mundo Novo × Catuaí in Campos Altos; MxC-CP: Mundo Novo × Catuaí in Capelinha; ICT-CA: Icatu in Campos Altos; ICT-CP: Icatu in Capelinha; MN-TP: Mundo Novo in Três Pontas; MN-CA: Mundo Novo in Campos Altos; MN-CP: Mundo Novo in Capelinha.

The standard deviation, which indicates the degree of data dispersion in relation to the mean, had the highest value estimated in the RT method (6.81). This suggested that in addition to overestimating the data and not considering compensation, this method was inappropriate for increasing dispersion around the mean (Table 4).

**Table 4.** Genetic parameter estimates obtained by different methods used to correct production data from F4 progenies obtained by crossing cultivars of Icatu × Catimor, Mundo Novo × Catuaí, and Icatu in the cities of Campos Altos (CA), Capelinha (C), and Três Pontas (TP).

| | Icatu × Catimor—CA | | | | Mundo Novo × Catuai—CA | | | | Mundo Novo × Catuai—C | | | |
|---|---|---|---|---|---|---|---|---|---|---|---|---|
| | *p*-Value | $\hat{r}_{\hat{g}g}$ | $r_G$ | IC | *p*-Value | $\hat{r}_{\hat{g}g}$ | $r_G$ | IC | *p*-Value | $\hat{r}_{\hat{g}g}$ | $r_G$ | IC |
| AA | 0.09 | 0.61 | 1.00 | 1.00 | 0.12 | 0.60 | 1.00 | 1.00 | 0.29 | 0.42 | 1.00 | 1.00 |
| RT | 0.23 | 0.46 | 0.66 | 0.14 | 0.26 | 0.46 | 0.86 | 0.73 | - | 0.00 | 0.00 | 0.00 |
| Zb | 0.18 | 0.52 | 0.81 | 0.71 | 0.21 | 0.51 | 0.92 | 0.73 | - | 0.00 | 0.00 | 0.00 |
| VC | 0.14 | 0.55 | 0.95 | 0.71 | 0.10 | 0.62 | 0.92 | 1.00 | 0.20 | 0.52 | 0.97 | 0.73 |
| ACI | 0.14 | 0.53 | 0.98 | 1.00 | 0.09 | 0.64 | 0.99 | 1.00 | 0.45 | 0.21 | 0.98 | 0.73 |
| ACA | 0.12 | 0.56 | 0.97 | 1.00 | 0.09 | 0.64 | 0.99 | 1.00 | 0.45 | 0.21 | 0.98 | 0.73 |
| Cr | 0.08 | 0.63 | 1.00 | 1.00 | 0.09 | 0.64 | 0.99 | 1.00 | 0.40 | 0.28 | 0.82 | 0.47 |
| | Mundo Novo × Catuai—TP | | | | Icatu—CA | | | | Icatu—C | | | |
| | *p*-Value | $\hat{r}_{\hat{g}g}$ | $r_G$ | IC | *p*-Value | $\hat{r}_{\hat{g}g}$ | $r_G$ | IC | *p*-Value | $\hat{r}_{\hat{g}g}$ | $r_G$ | IC |
| AA | 0.07 | 0.56 | 1.00 | 1.00 | 0.33 | 0.43 | 1.00 | 1.00 | 0.04 | 0.82 | 1.00 | 1.00 |
| RT | 0.06 | 0.58 | 0.93 | 0.00 | - | 0.00 | 0.00 | 0.00 | 0.19 | 0.59 | 0.59 | 0.29 |
| Zb | 0.06 | 0.57 | 0.96 | 0.00 | 0.27 | 0.27 | 0.65 | 0.53 | 0.14 | 0.65 | 0.71 | 0.29 |
| VC | 0.04 | 0.56 | 0.99 | 0.66 | - | 0.00 | 0.00 | 0.00 | 0.24 | 0.54 | 0.38 | 0.53 |
| ACI | 0.06 | 0.58 | 0.98 | 0.66 | 0.33 | 0.43 | 0.98 | 1.00 | 0.04 | 0.82 | 0.96 | 0.76 |
| ACA | 0.06 | 0.58 | 0.98 | 0.66 | 0.33 | 0.43 | 0.98 | 1.00 | 0.04 | 0.82 | 0.96 | 0.76 |
| Cr | 0.05 | 0.58 | 0.99 | 0.66 | 0.41 | 0.30 | 0.90 | 0.76 | 0.14 | 0.65 | 0.71 | 0.53 |
| | Mundo Novo—CA | | | | Mundo Novo—TP | | | | Mundo Novo—CA | | | |
| | *p*-Value | $\hat{r}_{\hat{g}g}$ | $r_G$ | IC | *p*-Value | $\hat{r}_{\hat{g}g}$ | $r_G$ | IC | *p*-Value | $\hat{r}_{\hat{g}g}$ | $r_G$ | IC |
| AA | 0.08 | 0.61 | 1.00 | 1.00 | 0.22 | 0.46 | 1.00 | 1.00 | 0.02 | 0.72 | 1.00 | 1.00 |
| RT | - | 0.00 | 0.00 | 0.00 | 0.19 | 0.48 | 0.81 | 0.38 | - | 0.00 | 0.00 | 0.00 |
| Zb | - | 0.00 | 0.00 | 0.00 | 0.18 | 0.49 | 0.90 | 0.69 | 0.34 | 0.33 | 0.79 | 0.69 |
| VC | 0.42 | 0.23 | 0.84 | 0.08 | 0.47 | 0.15 | 0.95 | 0.69 | 0.09 | 0.59 | 0.96 | 1.00 |
| ACI | 0.20 | 0.48 | 0.96 | 1.00 | 0.17 | 0.50 | 0.97 | 0.69 | 0.06 | 0.64 | 0.92 | 1.00 |
| ACA | 0.20 | 0.48 | 0.96 | 1.00 | 0.17 | 0.50 | 0.97 | 0.69 | 0.06 | 0.64 | 0.92 | 1.00 |
| Cr | 0.11 | 0.57 | 0.93 | 0.69 | 0.21 | 0.47 | 0.91 | 0.69 | 0.31 | 0.36 | 0.84 | 0.69 |

*p*-value: probability of significance of the genetic factor; $\hat{r}_{\hat{g}g}$: selective accuracy; $r_G$: genetic correlation; IC: index of coincidence; AA: absence of adjustment; RT: rule of three method; Zb: Zuber [5] method; VC: Vencovsky and Cruz [4] method; ACI: analysis of covariance based on the ideal stand; ACA: analysis of covariance based on the average stand; Cr: Cruz [*apud* 4] method.

For each set of adjusted data, the p-value, selective accuracy, and genetic correlation between progeny ordering and IC value of the five best progenies selected were estimated to evaluate the influence of plant loss on the selection used by coffee breeders in their routine work. The *p*-value was interpreted as a gradation of probabilities of accepting or rejecting H0 rather than being a fixed value above which no significant difference is considered. Therefore, regardless of the fixed level of significance, we analyzed the possibility of a change in the variation probability due to the analyzed factor.

Analysis of the F4 progenies generated by the Icatu × Catimor cross in Campos Altos showed that the lowest *p*-values were obtained with the Cr (0.08) and AA (0.09) methods, although they were not significant at the 5% level according to the standard procedure. These two methods (Cr and AA) were more accurate (0.63 and 0.61, respectively) and generated a 100% IC in the ACI and ACA methods (Table 4). Based on the IC values, the RT method resulted in the most discrepant transformation in the selections made (0.14). For this population, the best parameters were obtained using the Cr method, which resulted in greater selective accuracy and genetic correlation and an IC equal to the AA data, yielding a probability of differences due to genotype of 6.32%.

The progenies of Mundo Novo × Catuaí in Campos Altos obtained the best fit with the ACI, ACA, and Cr methods, which yielded the highest selective accuracy. Regarding the correlation and coincidence in selection, the ACI, ACA, and Cr methods again provided the same ordering and selection as the AA method. In Capelinha, the genetic variance was not

significant, indicating the difficulties in selecting superior genotypes in that location. The VC method performed well since it promoted the best genetic discrimination of progenies, as verified by the *p*-value. Thus, the correlation and IC values of the VC method were equal to those of the correction methods with the best fit (i.e., ACI and ACA), leading to the conclusion it may be the best data adjustment approach in this situation. Likewise, in Três Pontas, the VC method achieved greater discriminatory power and selected the same progenies as the methods with best fit (ACI and ACA) (Table 4).

For the Icatu progenies, whether in Campos Altos or Capelinha, the best parameter estimates were obtained with the ACI, ACA, and AA methods. There were no genetic variations among the progenies in Campos Altos, whereas genetic effects were highly significant among those in Capelinha, and the selective accuracy was high. The genetic correlation of the ACI and ACA corrected data with the AA data was high, generating a selection IC of 100% and 76% in Campos Altos and Capelinha, respectively.

In the Mundo Novo progenies, the ACI and ACA methods were distinguished in fitting the data to the model in the three experimental locations. In Campos Altos, the AA method presented the highest selective accuracy value. An identical pattern was observed for the progenies in Capelinha. By contrast, the AA, ACI, and ACA methods were not able to identify genetic differences in the progenies in Três Pontas and selected only 70% of coinciding progenies.

## 4. Discussion and Conclusion

Plot plant loss causes major problems in data analysis and experimental result interpretation, as it results in uneven stands and compromises experimental precision. Furthermore, this is especially problematic for perennial crops as they perpetuate errors throughout the duration of the experiment [13]. Currently, information regarding plant loss mediation in coffee plots and experiments in the field remain scarce. Therefore, in this study, an extensive EPAMIG database was analyzed to understand the influence of the compensatory effects of *Coffea arabica* progeny production on productivity data and identify alternatives to overcome this problem. In coffee studies, it is common practice to base inferences on four harvests, and these are considered sufficient for drawing meaningful conclusions [14,15]. Aiming to strengthen the robustness of these findings and provide a more comprehensive analysis of the crop's performance over time, we included data from six harvests.

Initially, the linear regression coefficient of productivity with the number of plants in the plot and the compensatory effect of the production of adjacent plants against an absent plant were estimated. This balance between production loss and compensation is genetically determined and depends on the plant spacing and arrangement failures on the plot [4]. Of the 54 harvests studied, 32 had a compensation coefficient value > 1, indicating a positive mean capacity for production compensation as a function of each failure present in the plot of 1.15 (Table 3). Considering that this set of data is quite representative of actual scenarios, coffee trees possibly have a significant capacity to compensate for the absence of plants in the plot or a decrease in the stand (in the case of wider spacing). These results are physiologically sound, and since compensation follows the natural productive increase in coffee trees, which reaches a peak between the sixth and eighth harvests, it stabilizes their biennial production until plant ageing [16].

The results obtained corroborate those from other researchers, who reported that plant production compensation in the same crop varies between experiments, demonstrating the influence of genotypes and soil and climate conditions. These effects were further verified in maize [17] and beans [18,19]. In this study, the observed differences in compensation for lack of competition (Table 3) can be attributed to the fact that coffee progenies vary in their ability to respond to spacing via root growth differentials [20] and respond to light via different plagiotropic branches, angles, and plant heights [21].

In this study, a mixed model was employed, considering the progeny factor as random and the block and harvest factors as fixed effects. The inclusion of genetic factors as random in the model is advocated in various studies due to the predictive nature of the estimators

derived from such an approach [22,23]. BLUP (Best Linear Unbiased Prediction) is widely acknowledged as one of the most robust estimators for handling random effects within a mixed model. In the context of plant breeding, the E-BLUP (Empirical Best Linear Unbiased Prediction) was applied, which involves estimating the BLUP from the sampled data. This method serves as the genotypic value, accounting for the proportion of phenotypic variance attributed to genetic effects while effectively removing irrelevant factors that do not significantly contribute to plant improvement. It is important to emphasize that in a balanced experiment (without missing data), the E-BLUP is equivalent to the progeny's phenotypic mean.

After verifying the fit of the model and analyzing whether variation is heritable and the phenotype is a faithful representation of the genotype, we observed the impact of data transformations in terms of plant loss in the plot on progeny selection. First, the analysis was performed in a split-plot design over time, which is a common procedure for production data in coffee breeding as it identifies precocious progenies in relation to production and allows genetic variance estimation free of progeny–crop and progeny–block interaction variances [15].

For progeny and cultivar competition tests, it is necessary to work on the data to verify their need for correction due to plant loss in the plots. Genetic correlation and IC analyses were performed to determine the magnitude of progeny ordering change and the proportion of similarity among the selections performed by the different methods in association with a probability chosen by the breeder. Here, the most adequate methods for correcting the productive data of the progeny tests were chosen based on the *p*-value and selective accuracy and verified for each population (Table 4). The results revealed that generalizations were not appropriate, making it impossible to establish fixed rules since the interaction of the evaluated plant with the cultivation environment results in a relatively plastic response.

The estimated compensation coefficients obtained from the various experiments varied greatly when the fixed factor of 0.3 used in the Zb method was applied, revealing this method to be ineffective. Only 13% of the studied estimates were similar to the 0.3 value recommended by Zuber [5], indicating the need to reconsider this correction method for coffee crops. In coffee cultivation, there is no linear proportionality between the number of plants and the observed yield. Thus, adjusting this trait based on the ideal stand through simple proportionality resulted in overestimated plant yields [24].

Likewise, stand correction using the RT method was also ineffective, as it did not provide greater estimates of selective accuracy in any of the experimental situations studied and instead presented the highest production means, which overestimated the mean production value. According to Vencovsky and Cruz [4], this method has been efficient in situations where the degree of failure compensation in the plot was low but was proven to be inefficient in cases where the compensation was high (close to 1). Although the RT method is not useful for data correction, it is necessary for calculating the compensation coefficient due to irregularities in the experimental stand.

An accurate method should consider both the decrease in production due to the presence of gaps and the increase obtained in neighboring plants due to production absence [25]. In this study, the covariance correction methods for average and ideal stands were efficient, as they promoted greater decrease in the residual variance via a reduced *p*-value and increased the experimental precision. Furthermore, it proved to be useful in separating the effect/interference of an unplanned variable (in this case, variation in plant stands) from another variable of interest [26]. These findings corroborated other researchers who analyzed maize [17] and grain sorghum [27].

In the search to establish a relationship between selection decisions based on data corrections and those based on the original data (unadjusted), we found that six of the nine selection decisions by the breeders were completely coincident with what was conducted in the populations of Icatu × Catimor [28,29], Mundo Novo × Catuaí [30,31], and Mundo Novo [32], Catuaí [33], and Icatu lineages [34]. Therefore, if the adjustments were

biologically correct, the breeders were correct 66.67% of the time. As breeding requires intensive resource use, a 33.33% probability of error in selection is considerably high and demonstrates, once again, the importance of considering the effect of plant loss in the plot. Recently, software tools have played a vital role in facilitating statistical analysis by streamlining complex procedures and enabling researchers to obtain accurate results. These tools, with their user-friendly interfaces and widespread availability, have made statistical analyses easier and more accessible, leading to improved efficiency and scientific rigor [26].

If we disregard the importance of properly applying a correction method in experiments with plot failures, the possible effects of competition between plots may result in genotype selection that have limited value for the trait and target environments. Furthermore, many of the actual genetic effects are small in magnitude for a polygenic trait, potentially contributing to statistical bias and error [35].

Despite not being the focus of this study, data correction should be understood in the context of the genotype–environment interaction. In this study, the data were analyzed according to experimentation locations within populations. However, after identifying the most suitable methods for correcting failures, these data can be used in joint analyses of the environments under study, increasing the chance of correct decisions being made.

Another aspect identified by the result interpretation of this study is the criterion for choosing the correction methods to be applied. This decision should collectively consider several parameters and first analyze the existence of genetic variance; selective accuracy; correlation between phenotype and genotype; and ordering of the progenies, lineages, or cultivars under evaluation. The comparison of progenies under equal conditions is essential to ensure the genetic progress of cultivars and the efficiency of breeding programs. Therefore, it is important to perform experiments that allow plants to express their genetic potential. Although many unforeseen events may occur, the experiments should be carefully conducted, especially in the crop formation phase.

Understanding and correcting plant losses in experimental plots is crucial to ensure data accuracy, avoid statistical bias, and select genotypes with appropriate genetic potential. These principles can be applied in agricultural experiments involving different crops, enabling the precise identification of promising genotypes and contributing to breeding program advancements.

**Author Contributions:** C.E.B.: Conceptualization, Data curation, Formal analysis, Investigation, Methodology, Validation, Visualization, Writing—original draft, Writing—review and editing. V.T.A.: Conceptualization, Data curation, Formal analysis, Investigation, Methodology. J.C.d.R.A.: Investigation, Methodology, Validation, Visualization, Writing—original draft, Writing—review and editing. F.M.A.G.: Conceptualization, Investigation, Methodology, Resources, Supervision, Validation, Visualization, Writing—original draft, Writing—review and editing. All authors have read and agreed to the published version of the manuscript.

**Funding:** This work was supported by Minas Gerais Agricultural Research Agency (EPAMIG). The authors would like to thank Fundação de Amparo à Pesquisa do Estado de Minas Gerais (FAPEMIG), Conselho Nacional de Desenvolvimento Científico e Tecnológico (CNPq), Consórcio de Pesquisa Café, and Instituto Nacional de Ciência e Tecnologia do Café (INCT-Café) for the financial support.

**Data Availability Statement:** Not applicable.

**Conflicts of Interest:** The authors declare no conflict of interest.

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
