# Peer review of "Missing Plants Effects and Stand Correction Methods in Coffea arabica Progeny Experiments"

_agronomy, doi:10.3390/agronomy13092374_

Round 1
Reviewer 1 Report
The manuscript has evaluated known methods for plant loss correction in coffee. They didn’t create a new method for the correction. It might be useful for coffee but with little significance to major crops such as maize and rice. Because the methods have long been discussed in biostatistics. And the experimental population is not very large, which might affect the result. I am afraid the manuscript is not enough for publication in Agronomy at the current form.
Minor concerns
The comma in tables should be dot.
The original data of all experiments should be listed in supplementary materials, which is necessary for readers who want to repeat the analysis.
The English is good. And a grammar checking is suggested.
Author Response
June 16, 2023
Dear Reviewer, 1,
We would like to express our sincere appreciation for the detailed comments and suggestions provided during the review of our manuscript titled “Best stand correction methods for attenuating the effects of plant loss in experimental plots of Coffea arabica progenies.” We have carefully considered all your observations and have diligently worked to improve the manuscript according to your recommendations.
First and foremost, we have addressed the concerns raised regarding the introduction. We have thoroughly revised the section, providing sufficient background information on the research topic, and including all relevant references. We have ensured that all cited references are appropriate and contribute to the theoretical framework of our study.
Furthermore, we have meticulously examined the research design to verify its suitability for the proposed objectives. We have reviewed and enhanced the description of the methods used, including details on the sample, data collection procedures, instruments, and statistical analyses. The research design is now aligned with the research objectives and provides sufficient information for study replication.
Regarding the results, we have made modifications to ensure their clear and comprehensible presentation. Additionally, the discussion has been carefully revised and rewritten to ensure proper support from the presented results.
With regards to your comment on the relevance of the methods used in our study, we acknowledge that these methods are already known and discussed in biostatistics. However, we believe that our research contributes to the field of agronomy, specifically in the context of coffee cultivation, by evaluating the application of these methods in a relevant setting. While we understand that there may be such studies for major crops such as corn and rice, we emphasize the significance of the study for coffee farming, highlighting the peculiarities and specific needs of this crop.
We have taken note of your observation regarding the size of the experimental population. Nevertheless, is important to note that the number of environments we utilized in our study, which was 11, is considered representative for coffee cultivation in our region. By incorporating a diverse range of environments, we aimed to capture the variability inherent in coffee production and ensure a comprehensive understanding of the crop's performance across different conditions. This approach enhances the generalizability of our findings and their applicability to the broader coffee farming community in our state. Furthermore, in the case of coffee studies, it is common practice to base inferences on four harvests, are considered sufficient for drawing meaningful conclusions. In our study, we went beyond this standard and included data from six harvests. By extending the observation period and incorporating additional harvests, we aimed to strengthen the robustness of our findings and provide a more comprehensive analysis of the crop's performance over time. While we acknowledge that a larger sample size may offer additional statistical power, it is important to balance practical considerations and the resources available for conducting research. Given the logistical constraints and resources at our disposal, we believe that our sample size of experimental plots adequately addresses the objectives of our study and provides valuable insights into coffee cultivation in our state.
Furthermore, we have replaced all commas with periods, as requested, in the tables presented in the manuscript. We appreciate this observation, as it contributes to the standardization and clarity of the data presented.
Regarding the original data from the experiments, we agree with your suggestion and have included all relevant details in the supplementary materials.
Once again, we sincerely thank you for your valuable feedback, which has greatly contributed to improving the quality of our manuscript. We believe that the revisions made address the concerns raised and substantially enhance the manuscript's suitability for publication in Agronomy Journal. We look forward to receiving further guidance or feedback from you during the final review process.
Thank you for your time and consideration.
Sincerely,
César Elias Botelho
Empresa de Pesquisa Agropecuária de Minas Gerais
Lavras, MG, Brasil
E-mail: cesarbotelho@epamig.br
Reviewer 2 Report
The justification of relevance in the introduction is insufficient.
Apparently, some parts of the formulas are missing in the explanation of data analysis methods (third page, third paragraph). Due to the lack of line numbering, it is difficult to specify the exact places with defects. It is better to present the entire section with methods of analysis in the form of well-readable formulas with an explanation for them.
For research methods of data analysis, in my opinion, it is desirable to use large samples, or to give a justification for the sample size (why do the authors believe that such a number of plants in the experiment is enough). To identify phenotypic differences, in my opinion, it is not enough to evaluate only one parameter, especially such a complex one as yield.
Author Response
June 20, 2023
Dear Reviewer, 2,
We sincerely appreciate your thorough evaluation and constructive feedback on our manuscript titled "Best Stand Correction Methods for Attenuating the Effects of Plant Loss in Experimental Plots of Coffea arabica Progenies" We have carefully considered all your comments and suggestions, and we have made significant improvements to address the concerns raised.
The introduction, methods, discussion, as well as the conclusions have been rewritten in response to the reviewer's comments. Regarding the introduction, we have revised it to provide sufficient background information and ensure the inclusion of all relevant references. We have worked diligently to strengthen the justification for the research's relevance, ensuring that it adequately highlights the importance of the study within the field. Additionally, the tables have been recreated, and relevant references have been included to enhance the overall quality and comprehensiveness of the article. These revisions aim to improve the clarity, accuracy, and scientific rigor of the manuscript, ensuring that it meets the standards set by the reviewer and aligns with the objectives and scope of the journal.
We acknowledge your observation regarding the missing parts of formulas in the explanation of the data analysis methods. In response, we have thoroughly revised this section. To enhance clarity and readability, we have now presented the entire section with well-defined and legible formulas. This modification ensures that readers can readily understand the analytical methods employed in our study.
We have taken note of your observation regarding the size of the experimental population. Nevertheless, is important to note that the number of environments we utilized in our study, which was 11, is considered representative for coffee cultivation in our region. By incorporating a diverse range of environments, we aimed to capture the variability inherent in coffee production and ensure a comprehensive understanding of the crop's performance across different conditions. This approach enhances the generalizability of our findings and their applicability to the broader coffee farming community in our state.
Furthermore, in the case of coffee studies, it is common practice to base inferences on four harvests, are considered sufficient for drawing meaningful conclusions. In our study, we went beyond this standard and included data from six harvests. By extending the observation period and incorporating additional harvests, we aimed to strengthen the robustness of our findings and provide a more comprehensive analysis of the crop's performance over time.
While we acknowledge that a larger sample size may offer additional statistical power, it is important to balance practical considerations and the resources available for conducting research. Given the logistical constraints and resources at our disposal, we believe that our sample size of experimental plots adequately addresses the objectives of our study and provides valuable insights into coffee cultivation in our state.
We have taken note of your comment regarding the lack of line numbering, which made it difficult to pinpoint exact locations with defects in the previous version. We used the journal template to prepare the manuscript, which is available on the individual journals’ Instructions for Authors pages.
We want to express our gratitude for your valuable insights and suggestions. We believe that the modifications and improvements made have significantly strengthened the manuscript. We are confident that the revised version now adequately addresses the concerns you raised, providing a more comprehensive and cohesive contribution to the field.
Thank you for your time and thorough review. We look forward to any additional guidance you may have during the final review process.
Sincerely,
César Elias Botelho
Empresa de Pesquisa Agropecuária de Minas Gerais
Lavras, MG, Brasil
E-mail: cesarbotelho@epamig.br
Reviewer 3 Report
The topic of the manuscript "Best Stand Correction Methods for Attenuating the Effects of Plant Loss in Experimental Plots of Coffea arabica Progenies " is timely and fits within the scope of the Journal. However, I have highlighted weaknesses in the manuscript that require the paper to be rewritten. These are descriptions of methods, equations, statistical parameters and the use of statistical methods. I provide detailed comments on the Materials and Methods chapter below. Without improving this chapter, the rest of the work makes no sense.
Detailed comments:
2. Materials and Methods
Text extract 1 - The compensatory effect was determined by estimating the compensation coefficient (a) using the estimator a=c/(Y Ì… ), where c is the linear regression coefficient proposed by Cruz [1] and Y Ì… is the mean production yield per plant obtained in the experiment.
Firstly, the number 1 in Cruz[1] is incorrect.
Secondly, nobody knows the linear regression coefficient proposed by Cruz. It is a doctoral thesis and it is not available. No one has ever described the linear regression coefficient proposed by Cruz. This should be done. This manuscript should be methodological.
Text extract 2 – paragraph „Following the recommendation by Steel et al. [9], … Zij represents the corrected yield and Yij the observed production in real plots/stands (Xij).”
Firstly, I would guess that Yij, Zij etc are variables and should be written in italics.
Secondly, How do the variables Yij, Zij etc. relate to Y Ì…?
Text extract 3 – paragraph „Analyses of the six harvests were carried out … using the square root of the heritability at the mean progeny level.”
Firstly, there is no model equation.
Secondly, why was the empirical best linear unbiased prediction (E-BLUP) score suddenly used?
The rest of the manuscript is irrelevant if there are no correctly written formulae, no explanation of the statistical methods used.
We do not know what is new compared to all the previous methods cited?
Author Response
June 20, 2023
Dear Reviewer, 3,
Thank you for your detailed feedback on our manuscript titled "Best Stand Correction Methods for Attenuating the Effects of Plant Loss in Experimental Plots of Coffea arabica Progenies". We appreciate your assessment of the weaknesses in our manuscript and agree that the article requires significant revision.
The introduction, methods, discussion, as well as the conclusions have been rewritten in response to the reviewer's comments. We have thoroughly revised the manuscript to address the concerns raised. Additionally, the tables have been recreated, and relevant references have been included to enhance the overall quality and comprehensiveness of the article. These revisions aim to improve the clarity, accuracy, and scientific rigor of the manuscript, ensuring that it meets the standards set by the reviewer and aligns with the objectives and scope of the journal.
We have taken note of your observation regarding the size of the experimental population. Nevertheless, is important to note that the number of environments we utilized in our study, which was 11, is considered representative for coffee cultivation in our region. By incorporating a diverse range of environments, we aimed to capture the variability inherent in coffee production and ensure a comprehensive understanding of the crop's performance across different conditions. This approach enhances the generalizability of our findings and their applicability to the broader coffee farming community in our state. Furthermore, in the case of coffee studies, it is common practice to base inferences on four harvests, are considered sufficient for drawing meaningful conclusions. In our study, we went beyond this standard and included data from six harvests. By extending the observation period and incorporating additional harvests, we aimed to strengthen the robustness of our findings and provide a more comprehensive analysis of the crop's performance over time. While we acknowledge that a larger sample size may offer additional statistical power, it is important to balance practical considerations and the resources available for conducting research. Given the logistical constraints and resources at our disposal, we believe that our sample size of experimental plots adequately addresses the objectives of our study and provides valuable insights into coffee cultivation in our state.
We have addressed your comments and made the necessary improvements to the Materials and Methods section, as outlined below:
- The reference to Cruz [1] has been corrected. We apologize for the error in citing the reference number, and it has been rectified accordingly.
- Regarding the coefficient of linear regression proposed by Cruz, we understand your concern that it is not widely known or described in the literature. Therefore, we have replaced the reference to Cruz (1971) with Vencovsky and Cruz (1991), which provides a more appropriate and well-established source for the methodology.
- We have revised the formatting of variables such as Yij and Zij to be written in italics, as per your suggestion.
- Regarding the relationship between the variables Yij, Zij, and the mean Ȳ, we would like to provide a clarification. Yij is the yield corrected using the rule of three, taking into account the number of failures in the plot. On the other hand, Zij represents the adjusted yield, where the correction is applied based on the variable stand. The mean Ȳ is not the original mean but rather a representation of the overall average considering the adjusted yields. The application of the correction factors aims to account for the impact of plant loss in the plot and provide a more accurate assessment of the productivity data. Therefore, Yij is adjusted proportionally to the number of failures, while Zij reflects the corrected yield, and Ȳ represents the adjusted mean value derived from the corrected yields.
- Regarding Text extract 3, the paragraph ‘Analyses of the six harvests were carried out … using the square root of the heritability at the mean progeny level’ We acknowledge the absence of an explicit equation model in the previous version of the manuscript. To address this, we have now included the relevant equation model in the corresponding section, providing a clear understanding of the statistical methodology employed.
- The empirical best linear unbiased prediction (EBLUP) was used because when the components of (co)variance (G and R) or their estimates are unknown. When the matrices G and R are unknown, with only estimates obtained through some method, it is appropriate to use EBLUE (empirical best linear unbiased estimator) and EBLUP (empirical best linear unbiased predictor) instead of BLUE and BLUP. The addition of the term "empirical" signifies this type of approximation, addressing the issue raised in your comment.
- We have revised and clarified the formulas and methods to ensure their accurate representation in the revised version of the manuscript.
The new contributions of the article include addressing the compensatory effect of crop under plot failure, specifically in Coffea plants, and how it affects experimental accuracy in coffee crop experiments. The article highlights the scarcity of information on dealing with plant losses in coffee plot experiments and presents an extensive analysis of an EPAMIG database to understand the influence of the compensatory effect on productivity data. Additionally, the study explores the relationship between selection decisions based on data corrections and those based on the original unadjusted data, revealing a significant probability of error in selection decisions by breeders. These findings emphasize the importance of considering the effect of plant loss in the plot and provide insights and alternatives for overcoming this problem in coffee crop experiments.
We would like to express our gratitude for your thorough evaluation of our manuscript and your constructive comments. We have taken them into account and made the necessary revisions to improve the clarity and scientific rigor of our work. We hope that these revisions adequately address your concerns, and we remain open to any further suggestions or inquiries you may have.
Sincerely,
César Elias Botelho
Empresa de Pesquisa Agropecuária de Minas Gerais
Lavras, MG, Brasil
E-mail: cesarbotelho@epamig.br
Round 2
Reviewer 1 Report
The authors have addressed most of the comments. The manuscript could be accepted after grammar checking.
A grammar checking is necessary before publication.
Author Response
July 27, 2023
Dear Reviewer, 1,
We would like to express our sincere gratitude for the time and effort you invested in reviewing our work. Furthermore, we would like to inform you that the English text underwent meticulous revision by the renowned company Editage. We acknowledge Editage's high standard of excellence in their revisions, and as a result, we opted for their services to ensure our work adheres to international linguistic quality standards.
If you have any questions or require further information, please do not hesitate to contact us.
Sincerely,
César Elias Botelho
Empresa de Pesquisa Agropecuária de Minas Gerais
Lavras, MG, Brasil
E-mail: cesarbotelho@epamig.br
Reviewer 3 Report
The topic of the manuscript 'Best Stand Correction Methods for Attenuating the Effects of Plant Loss in Experimental Plots of Coffea arabica Progenies' is timely and within the scope of the journal. However, I have highlighted shortcomings in the manuscript that require a rewrite of the article. Below, I provide detailed comments on Chapters: 3.Materials and Methods and 4.Results. Without correcting these chapters, the rest of the paper makes no sense.
Detailed comments:
2. Materials and Methods
Text extract 1 - The compensatory effect was determined by estimating the compensation coefficient (a) using the estimator a=c/(Y Ì… ), where c is the linear regression coefficient proposed by Cruz [1] and Y Ì… is the mean production yield per plant obtained in the experiment.
Firstly, the number 1 in Cruz[1] is incorrect.
Secondly, nobody knows the linear regression coefficient proposed by Cruz. It is a doctoral thesis and it is not available. No one has ever described the linear regression coefficient proposed by Cruz. This should be done. This manuscript should be methodological.
[This comment of mine has been corrected.]
Text extract 2 – paragraph „Following the recommendation by Steel et al. [9], … Zij represents the corrected yield and Yij the observed production in real plots/stands (Xij).”
Firstly, I would guess that Yij, Zij etc are variables and should be written in italics.
Secondly, How do the variables Yij, Zij etc. relate to Y Ì…?
[This comment of mine has been corrected.]
Text extract 3 – paragraph „Analyses of the six harvests were carried out … using the square root of the heritability at the mean progeny level.”
Firstly, there is no model equation.
Secondly, why was the empirical best linear unbiased prediction (E-BLUP) score suddenly used?
The rest of the manuscript is irrelevant if there are no correctly written formulae, no explanation of the statistical methods used.
We do not know what is new compared to all the previous methods cited?
Now we have a split-plot model, but it is wrong.
A split-plot model has at least two errors and here there is only one.
Unfortunately, there are no ANOVA results in the manuscript and I cannot say whether the model is badly written or whether the experiment was badly planned and conducted.
In this work we need to show the methodology. We need good models and results that can be as an appendix. This will allow readers to understand the research method and how they should analyse their data.
Further on, we have a lot of unknowns.
3. Results
Text extract 4 – paragraph „Analysis of variance testing of the number of missing plants in each plot revealed whether they had occurred by chance (Table 2).”
As I wrote earlier, unfortunately there are no ANOVA results in the manuscript and we cannot write conclusions without showing the reader the results.
Text extract 5 – paragraph „Table 2. List of experiments, sites, and compensation coefficient (a) values for experiments and har-vests from 1 to 6 (H 01 to H 06).”
The description of Table 2 is wrong. It is not clear what has been presented.
The rest of the manuscript is irrelevant unless the split-plot model is corrected with a full description of the experiment carried out.
As we know, in a split-plot the order in which the factors are set is very important. We do not have this information in the manuscript.
We need the ANOVA results, without them we cannot verify the written conclusions.
Author Response
July 27, 2023
Dear Reviewer 3,
We would like to express our sincere gratitude for taking the time to review our work and providing us with a wealth of constructive feedback. Your valuable suggestions have significantly elevated the quality of our manuscript, and we are truly grateful for your expertise and insights.
We would like to extend our heartfelt thanks for bringing up the issue regarding the proposed linear regression coefficient by Cruz. As you rightly pointed out, the original thesis is currently unavailable, making it challenging to reference this specific coefficient. In this way, we replaced the citation of the reference in the text. As an alternative, we considered that presenting a methodological manuscript could serve as a useful approach. However, we must clarify that our primary aim in this study was to identify and address a practical research problem related to mitigating the impact of randomly missing plants. Our findings revealed the existence of compensatory effects in coffee experiments, highlighting the importance of appropriately correcting grain yield data. In our view, this work holds theoretical and practical value for plant breeding. Future research may delve into detailed statistical frameworks and propose novel methodologies.
We have carefully incorporated the formulas for accuracy and heritability, as per your suggestion. We acknowledge the importance of E-BLUP (Empirical Best Linear Unbiased Prediction) as an estimator for random factors in a mixed model. As demonstrated in the text, we considered the progeny factor as random and the block and harvest factors as fixed effects, thereby generating a mixed model. BLUP has been widely recognized for its reliability in handling random effects, and E-BLUP, derived from the sampled data, serves as a valuable genotypic value, accounting for genetic effects while eliminating irrelevant factors in plant breeding, as supported by various studies (Smith et al., 2007; Duarte and Resende, 2007; Piepho, 2008). Additionally, we have included a mention of the significance of E-BLUP in a balanced experiment, where it is equivalent to the progeny's phenotypic mean. This information has been thoroughly added to the discussion.
With respect to the use of the split-plot design, we appreciate your query on this matter. The model we employed is indeed a split-plot in time, as thoroughly explained in the manuscript. The primary factor is the progeny, and the secondary factor is the harvest. The first error in this case arises from the progeny-block interaction. We made this information clearer in the text. It is essential to note that the primary distinction between this model and a classic split-plot design lies in the non-randomization of plots across harvests, leading to covariance between measurements. We considered a suitable compound symmetry for this covariance structure, tailored to the split-plot model in time.
Regarding the absence of ANOVA results in the manuscript, we want to clarify that we conducted ANOVA to ascertain which dataset could be corrected using the methods employed. Specifically, we aimed to discern whether the lack of plants was caused by genotypes or occurred randomly. We also performed ANOVA for fixed effects and variance component estimation for random effects across all experiments, and these data are being sent in attachment. However, due to the reasons previously mentioned, we decided not to include these data in the article. Instead, we presented the key estimated parameters, such as p-values and accuracy, to validate our conclusions.
In light of your valuable feedback, we have reworked the title of Table 2 and sincerely apologize for the errors that occurred during the manuscript preparation.
All modifications made to the text are highlighted in blue colour.
Once again, we express our deep appreciation for your time, expertise, and dedication in reviewing our work. Your insights have been instrumental in improving the quality of our research, and we are grateful for your invaluable contribution. We are confident that the revised version now adequately addresses the concerns you raised, providing a more comprehensive and cohesive contribution to the field.
Sincerely,
Cesar Elias Botelho
Empresa de Pesquisa Agropecuária de Minas Gerais
Lavras, MG, Brazil
Email: cesarbotelho@epamig.br